# McDiarmid-Type Inequalities for Graph-Dependent Variables and Stability Bounds

**Rui (Ray) Zhang** [*]
School of Mathematics
Monash University
rui.zhang@monash.edu

**Xingwu Liu** [†]
Institute of Computing Technology,
Chinese Academy of Sciences.
University of Chinese Academy of Sciences
liuxingwu@ict.ac.cn

**Yuyi Wang**
ETH Zurich, Switzerland
X-Order Lab, China
yuyiwang920@gmail.com

**Liwei Wang**
Key Laboratory of Machine Perception, MOE,
School of EECS, Peking University
Center for Data Science, Peking University
wanglw@cis.pku.edu.cn

## Abstract

A crucial assumption in most statistical learning theory is that samples are independently and identically distributed (i.i.d.). However, for many real applications, the i.i.d. assumption does not hold. We consider learning problems in which examples are dependent and their dependency relation is characterized by a graph. To establish algorithm-dependent generalization theory for learning with non-i.i.d. data, we first prove novel McDiarmid-type concentration inequalities for Lipschitz functions of graph-dependent random variables. We show that concentration relies on the forest complexity of the graph, which characterizes the strength of the dependency. We demonstrate that for many types of dependent data, the forest complexity is small and thus implies good concentration. Based on our new inequalities, we establish stability bounds for learning graph-dependent data.

## 1 Introduction

Generalization theory is at the foundation of machine learning. It quantifies how accurate a model would predict on the test data which the learning algorithm is not able to access during training. It usually relies on a crucial assumption: The data are independently and identically distributed (i.i.d.). The i.i.d. assumption allows one to use many powerful tools from probability to prove strong generalization error bounds. However, in real applications, the data are often non-i.i.d. i.e., the data collected can be dependent. There have been extensive discussions on why and how the data are dependent. We refer the readers to [1, 2].

Establishing generalization theory for dependent data has received a lot of attention [3, 4, 5, 6, 7]. A major line of research in this direction models the data dependency by various types of mixing such as $\alpha$-mixing [8], $\beta$-mixing [9], $\phi$-mixing [10], $\eta$-mixing [11], etc. Mixing models have been used in statistical learning theory to establish generalization error bounds based on Rademacher complexity [4, 6, 12] or algorithmic stability [3, 12, 13] via concentration results [14] or independent

---

[*]This work was done when this author was a master student at the Institute of Computing Technology, Chinese Academy of Sciences and University of Chinese Academy of Sciences. This research forms part of Rui (Ray) Zhang's master thesis submitted to the University of Chinese Academy of Sciences in May 2019.

[†]Corresponding author

blocking technique [15]. In these models, the mixing coefficients measure the extent to which the data are dependent to each other. Similar to the mixing models, learning under Dobrushin's condition [16] is also investigated via concentration results [17, 18, 19] using Dobrushin's interaction matrix [20]. Although the results under the various mixing conditions and Dobrushin's condition are fruitful, they are faced with difficulties in application: It is sometimes difficult to determine the quantitative dependency among data points. On the other hand, determining whether two data are dependent or not is often much easier. In this paper, we focus on such qualitative dependency of data. We use simple graphs as a natural tool to describe the dependency among data, and establish generalization theory for such graph-dependent data.

A basic building block of generalization theory is concentration inequality. Different settings and different assumptions require different concentration tools. The less we assume, the more powerful tools we need. In order to establish generalization theory for dependent data, standard concentration for i.i.d. data no longer applies. One must develop concentration inequalities for dependent data, which is a very challenging task.

In his seminal work [21], Janson proved an elegant concentration inequality for graph-dependent data. The inequality is a beautiful extension of Hoeffding inequality. It bounds the probability that the summation of graph-dependent random variables deviates from its expected value, in terms of the fractional coloring number of the dependency graph. Janson's inquality has been extended to any functions that can be decomposed into the summation of some functions of independent random variables [22]. This extension enables to establish generalization error bounds for graph-dependent data via fractional Rademacher complexity.

In [5], PAC-Bayes bounds for classification with non-i.i.d. data are obtained based on fractional colorings of graphs. The results also hold for specific learning settings such as ranking and learning from stationary $\beta$-mixing distributions. In [23], Ralaivola and Amini established new concentration inequalities for fractionally sub-additive and fractionally self-bounding functions of dependent variables. Their results are based on the fractional chromatic numbes and the entropy method. In [24, 25], Wang et al. used hypergraphs to model dependent random variables that are generated by independent ones. Leveraging the notion of fractional matching, they also establish concentration inequalities of Hoeffding- or Bernstein-type.

Though fundamental and elegant, the above generalization bounds are algorithm-independent. They considered the complexity of the hypothesis space and data distribution, but does not involve the learning algorithm. To derive better generalization bounds, there are growing interests in developing algorithm-dependent generalization theories. This line of research heavily relies on the algorithmic stability. A key advantage of stability bounds is that they are tailored to specific learning algorithms, exploiting their particular properties.

How can we establish algorithmic stability theory for graph-dependent data? Note that under the assumption of i.i.d. data, Hoeffding-type concentration inequality, which bounds the deviation of sample average from expectation, is not strong enough to prove stability-based generalization. On the contrary, McDiarmids inequality characterizes the concentration of general Lipschitz functions of i.i.d. random variables, hence serving as the key tool for proving the stability theory. Therefore, to build algorithmic stability theory for non-i.i.d. samples, one has to develop McDiarmid-type concentration for graph-dependent random variables.

In this paper, we prove the first McDiarmid-type concentration inequality for graph-dependent random variables in terms of a new notion called forest complexity, which measures the strength of the dependency. It turns out that for various dependency graphs, it is easy to estimate the forest complexity. The proposed concentration inequality enables us to prove stability-based generalization bounds for graph-dependent data. Our results provide basic tools for understanding learning with overparameterized models.

The rest of the paper is organized as follows. In section 2, we briefly introduce the notations and related results. In section 3, we establish McDiarmid-type inequalities for acyclic dependency graphs, and extend the concentration results to the general dependency graphs. In section 4, we apply our concentration results to the learning theory and establish generalization error bounds for learning graph-dependent data via algorithmic stability, we also provide an application of learning $m$-dependent data. Section 5 concludes the paper and points out the future research directions.

The supplementary materials can be found in [26].

## 2 Preliminaries

In this section, we present the notations and the basic McDiarmid's inequality for i.i.d. random variables.

Throughout this paper, let $n$ be a positive integer with $[n]$ standing for the set $\{1, 2, \ldots, n\}$. Let $\Omega_i$ be a Polish space for any $i \in [n]$, $\Omega = \prod_{i \in [n]} \Omega_i$ be the product space, $\mathbb{R}$ be the set of real numbers, $\mathbb{R}_+$ be the set of non-negative real numbers, $\mathbb{N}_+$ be the set of non-negative integers.

Concentration inequalities are fundamental tools in statistical learning theory. They are essentially tail probability bounds indicating how much a function of random variables deviates from some value that is usually the expectation. Among the most powerful ones is the McDiarmid's inequality which establishes a sharp, even tight in some cases, bound on the concentration, when the function satisfies $\mathbf{c}$-Lipschitz condition (bounded differences condition), namely, does not depend too much on any individual variable.

**Definition 2.1** (**c**-Lipschitz). *Given a vector* $\mathbf{c} = (c_1, \ldots, c_n) \in \mathbb{R}_+^n$, *a function* $f : \Omega \to \mathbb{R}$ *is said to be* $\mathbf{c}$*-Lipschitz if for any* $\mathbf{x} = (x_1, \ldots, x_n), \mathbf{x}' = (x_1', \ldots, x_n') \in \Omega$, *it satisfies*

$$|f(\mathbf{x}) - f(\mathbf{x}')| \leq \sum_{i=1}^{n} c_i \mathbf{1}_{\{x_i \neq x_i'\}},$$

*where* $c_i$ *is called the* $i$*-th Lipschitz coefficient of* $f$.

**Theorem 2.2** (McDiarmid's inequality [27]). *Suppose* $f : \Omega \to \mathbb{R}$ *is* $\mathbf{c}$*-Lipschitz, and* $\mathbf{X} = (X_1, \ldots, X_n)$ *is a vector of independent random variables with each* $X_i$ *taking values in* $\Omega_i$. *Then for any* $t > 0$, *the tail probability satisfies*

$$\mathbf{Pr}\left(f(\mathbf{X}) - \mathbf{E}[f(\mathbf{X})] \geq t\right) \leq \exp\left(-\frac{2t^2}{\|\mathbf{c}\|_2^2}\right). \tag{1}$$

Notice that the McDiarmid's inequality works for independent random variables. Janson's Hoeffding-type inequality [21] for graph-dependent random variables is a special case of McDiarmid-type inequality when the function is a summation. Specifically, when $f(\mathbf{X}) = \sum_{i=1}^{n} X_i$ with each $X_i$ ranging over an interval of length $c_i$,

$$\mathbf{Pr}\left(\sum_{i=1}^{n} X_i - \mathbf{E}\left[\sum_{i=1}^{n} X_i\right] \geq t\right) \leq \exp\left(-\frac{2t^2}{\chi^*(G)\|\mathbf{c}\|_2^2}\right), \tag{2}$$

where $\mathbf{c} = (c_1, \ldots, c_n)$ and $\chi^*(G)$ is the fractional coloring number of a dependency graph $G$ of random variables $\mathbf{X}$.

## 3 McDiarmid Concentration for Graph-dependent Random Variables

In this section we present our first set of main results, the McDiarmid-type concentration inequalities (i.e., concentration of Lipschitz functions) for graph-dependent random variables. The results in this section will serve as the tools for developing learning theory for dependent data.

We start from the simplest case that the dependency graph is acyclic, i.e., trees or forests. We prove McDiarmid-type concentration bounds for trees and forests with very simple forms. These inequalities are then extended to general graphs. To this end, we introduce the notion of forest complexity, which characterizes to what extent a general graph can be best approximated by a forest. We prove McDiarmid-type concentration inequality for general graph-dependent random variables in terms of the forest complexity. Finally we demonstrate that for many important classes of graphs, forest complexity is easy to estimate.

Below we first define the notion of dependency graphs, which is a widely used model in probability, statistics, and combinatorics, see [28, 29, 30, 31, 32] for examples.

**Definition 3.1** (Dependency Graphs). *An undirected graph* $G$ *is called a dependency graph of a random vector* $\mathbf{X} = (X_1, \ldots, X_n)$ *if*

1. $V(G) = [n]$

2. *if* $I, J \subset [n]$ *are non-adjacent in* $G$, *then* $\{X_i\}_{i \in I}$ *and* $\{X_j\}_{j \in J}$ *are independent.*

## 3.1 McDiarmid Concentration for Acyclic Graph-dependent Variables

Our first result is for the case that the dependency graph is a tree.

**Theorem 3.2.** *Suppose that $f : \Omega \to \mathbb{R}$ is a $\mathbf{c}$-Lipschitz function and $G$ is a dependency graph of a random vector $\mathbf{X}$ that takes values in $\Omega$. If $G$ is a tree, then for any $t > 0$, the following inequality holds:*

$$\mathbf{Pr}(f(\mathbf{X}) - \mathbf{E}[f(\mathbf{X})] \geq t) \leq \exp\left(-\frac{2t^2}{\sum_{\langle i,j \rangle \in E(G)}(c_i + c_j)^2 + c_{\min}^2}\right), \tag{3}$$

*where $c_{\min}$ is the minimum entry in $\mathbf{c}$.*

The proof of this theorem relies on decomposing $f(\mathbf{X}) - \mathbf{E}[f(\mathbf{X})]$ into the summation $\sum_{i=1}^{n} V_i$ with $V_i := \mathbf{E}[f(\mathbf{X})|X_1, \dots X_i] - \mathbf{E}[f(\mathbf{X})|X_1, \dots X_{i-1}]$. We show that each $V_i$ ranges in an interval of length at most $c_i + c_j$, where $j$ is the parent of $i$ in the tree (in the proof, we make the tree rooted by choosing the vertex with the minimum Lipschitz coefficient as the root). The theorem is then proved by applying the Chernoff-Cramér technique to $\sum_{i=1}^{n} V_i$. For details, please refer to Subsection A.1 in the supplementary materials.

Like McDiarmid's inequality, Theorem 3.2 also claims a deviation probability bound that decays exponentially. The decay rate is determined by two interplaying factors. One is the Lipschitz coefficient that is inherent to the function. The other is the pattern of the dependency, namely, which random variables are dependent and connected by an edge.

We then generalize the above result to the case where dependency graph $G$ is a forest.

**Theorem 3.3.** *Suppose that $f : \Omega \to \mathbb{R}$ is a $\mathbf{c}$-Lipschitz function and $G$ is a dependency graph of a random vector $\mathbf{X}$ that takes values in $\Omega$. If $G$ is a forest consisting of trees $\{T_i\}_{i \in [k]}$, then for any $t > 0$, the following inequality holds:*

$$\mathbf{Pr}(f(\mathbf{X}) - \mathbf{E}[f(\mathbf{X})] \geq t) \leq \exp\left(-\frac{2t^2}{\sum_{\langle i,j \rangle \in E(G)}(c_i + c_j)^2 + \sum_{i=1}^{k} c_{\min,i}^2}\right), \tag{4}$$

*where $c_{\min,i} = \min\{c_j : j \in V(T_i)\}$.*

Theorem 3.3 can be proved in a similar way as Theorem 3.2. The detailed proof is presented in Subsection A.2 of the supplementary materials.

We point out that Theorem 3.3 is a strict generalization of the McDiarmid's inequality for i.i.d. random variables. If all the random variables are independent, i.e., there is no edge in the dependency graph, then it is clear that Eq. (4) degenerates exactly to Eq. (1).

Theorem 3.3 also clearly demonstrates how dependency between random variables affects concentration. The decay rate of the probability that $f(\mathbf{X})$ deviates from its expectation is approximately reversely proportional to the number of edges in the dependency graph.

## 3.2 McDiarmid Concentration for General Graphs

In this subsection, we consider general graphs. Our basic idea for handling general graphs is to use a forest to approximate the graph. Specifically, we partition the variables into groups so that the dependency graph of these groups is a forest. We try to find the optimal forest approximation, which leads to the notion of forest complexity. We then prove McDiarmid-type concentration inequality for general graph-dependent random variables in terms of its forest complexity, which yields a very simple form.

We first define the concept of forest approximation.

**Definition 3.4** (Forest Approximation)**.** *Given a graph $G$, a forest $F$, and a mapping $\phi : V(G) \to V(F)$, if $\phi(u) = \phi(v)$ or $\langle \phi(u), \phi(v) \rangle \in E(F)$ for any $\langle u, v \rangle \in E(G)$, we say that $(\phi, F)$ is a forest approximation of $G$. Let $\Phi(G)$ denote the set of forest approximations of $G$.*

Intuitively, a forest approximation is transforming a graph into a forest by merging vertices and removing the incurred self-loops and multi-edges. In this way, we rule out the redundant variables that heavily depend on others and thus contribute little to concentration.

Based on forest approximation, we define the notion of forest complexity of a graph, which intuitively measures how much the graph looks like a forest.

**Definition 3.5** (Forest Complexity). *Given a graph $G$ and any forest approximation $(\phi, F) \in \Phi(G)$ with $F$ consisting of trees $\{T_i\}_{i \in [k]}$, let*

$$\lambda_{(\phi,F)} = \sum_{\langle u,v \rangle \in E(F)} \left( |\phi^{-1}(u)| + |\phi^{-1}(v)| \right)^2 + \sum_{i=1}^{k} \min_{u \in V(T_i)} |\phi^{-1}(u)|^2.$$

*We call*

$$\Lambda(G) = \min_{(\phi,F) \in \Phi(G)} \lambda_{(\phi,F)}$$

*the forest complexity of the graph $G$.*

Now we are ready to state our McDiarmid-type concentration inequality for general graph-dependent random variables.

**Theorem 3.6.** *Suppose that $f : \mathbf{\Omega} \to \mathbb{R}$ is a $\mathbf{c}$-Lipschitz function and $G$ is a dependency graph of a random vector $\mathbf{X}$ that takes values in $\mathbf{\Omega}$. For any $t > 0$, the following inequality holds:*

$$\mathbf{Pr}(f(\mathbf{X}) - \mathbf{E}[f(\mathbf{X})] \geq t) \leq \exp\left( -\frac{2t^2}{\Lambda(G)\|\mathbf{c}\|_\infty^2} \right).$$

With the tool of forest approximation, we reduce the concentration problem defined on graphs to that defined on forests. Basically, we use a new variable to represent each set of the original variables that are merged together by the forest approximation. The function can be equivalently transformed into a function of the new variables whose dependency graph is the forest. The proof is done by applying Theorem 3.3 to the new function. For details, please refer to Subsection A.3 in the supplementary materials.

Like the above theorems, Theorem 3.6 also establishes an exponentially decaying probability of deviation. The decay rate is totally determined by the Lipschitz coefficient of the function and the forest complexity of the variables' dependency graph. Intuitively, the more the dependency graph looks like a forest, the faster the deviation probability decays. This uncovers how the dependencies among random variables influence concentration.

## 3.3 Illustrations and Examples

This subsection consists of two parts. In the first part we review a widely-studied random process that generates dependent data whose dependency graph can be naturally constructed. In the second part, we deal with some dependency graphs to show that in many cases, the forest complexity is small and easy to estimate.

Consider a data generating procedure modeled by the *spatial Poisson point process*, which is a Poisson point process on $\mathbb{R}^2$ (See [33, 34] for discussions of using this process to model data collection in various machine learning applications.) The number of points in each finite region follows a Poisson distribution, and the number of points in disjoint regions are independent. Given a finite set $\mathcal{I} = \{I_i\}_{i=1}^{n}$ of regions in $\mathbb{R}^2$, let $X_i$ be the number of points in region $I_i$, $1 \leq i \leq n$. Then the graph $G\left([n], \{\langle i,j \rangle : I_i \cap I_j \neq \emptyset\}\right)$ is a dependency graph of the random variables $\{X_i\}_{i=1}^{n}$.

We present three examples to demonstrate that estimating the forest complexity $\Lambda(G)$ is usually easy. All the examples can naturally appear in the above process.

**Example 3.7** ($G$ is a tree). In this case, the identity map between $G$ and itself is a forest approximation of $G$. Then $\Lambda(G) \leq |E(G)|(1+1)^2 + 1 = 4n - 3 = O(n)$. We get an upper bound of $\Lambda(G)$ that is linear in the number of variables, which is almost tight compared with Hoeffding's inequality or Janson's result (see (2) with $\chi^*(G) = 2$).

**Example 3.8** ($G$ is a cycle $C_n$). If $n$ is even, a forest approximation is illustrated in Figure 1, where the cycle is approximated by a path $F$ of length $\frac{n}{2}$. The approximation $\phi$ maps any vertex of $G$ to the vertex of $F$ having the same shape, so each gray belt stands for a preimage set of $\phi$. We will keep this convention in the rest of this section. By the illustrated forest approximation,

$\Lambda(G) \leq 2 \times (1+2)^2 + (\frac{n}{2} - 2)(2+2)^2 + 1 = 8n - 13 = O(n)$. When $n$ is odd, according to the forest approximation shown in Figure 2, $\Lambda(G) \leq (1+2)^2 + (\frac{n-1}{2} - 1)(2+2)^2 + 1 = 8n - 14 = O(n)$. Since $\chi^*(G)$ is 2 or 3, our bound is again very tight compared with Jansons result.

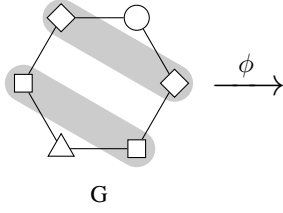 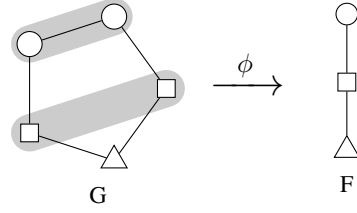

Figure 1: A forest approximation of $C_6$      Figure 2: A forest approximation of $C_5$

**Example 3.9** ($G$ is a grid). Suppose $G$ is a two-dimensional $(m \times m)$-grid. Then $n = m^2$. Considering the forest approximation illustrated in Figure 3, $\Lambda(G) \leq 2[3^2 + 5^2 + \ldots + (2m-1)^2] + 1 = \frac{2m(2m+1)(2m-1)-3}{3} = O(m^3) = O(n^{\frac{3}{2}})$

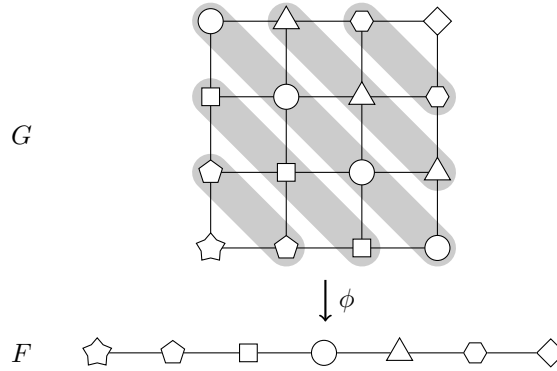

Figure 3: A forest approximation of the $(4 \times 4)$-gird

# 4 Generalization Theory for Learning from Graph-dependent Data

This section establishes stability generalization error bounds for learning from graph-dependent data, using the concentration inequalities derived in the last section.

Consider the supervised learning setting: Let $\mathbf{S} = ((x_1, y_1), \ldots, (x_n, y_n)) \in (\mathcal{X} \times \mathcal{Y})^n$ be a training sample of size $n$, where $\mathcal{X}$ is the input space and $\mathcal{Y}$ is the output space. Let $D$ be the underlying distribution of data on $\mathcal{X} \times \mathcal{Y}$. Assume that all the training data points $(x_i, y_i)$'s have the same marginal distribution $D$ and that $G$ is a dependency graph of $\mathbf{S}$.

Throughout this section, fix a non-negative loss function $\ell : \mathcal{Y} \times \mathcal{Y} \to \mathbb{R}$. For any hypothesis $f : \mathcal{X} \to \mathcal{Y}$, the empirical error on sample $\mathbf{S}$ is

$$\widehat{R}(f) = \frac{1}{n} \sum_{i=1}^{n} \ell(y_i, f(x_i)).$$

For learning from dependent data, the generalization error can be defined in various ways. We adopt the following widely-used one [35, 36, 37, 38]

$$R(f) = \mathbf{E}_{(x,y) \sim D}[\ell(y, f(x))], \tag{5}$$

which assumes that the test set is independent of the training set.

## 4.1 Bounding Generalization Error via Algorithmic Stability

Algorithmic stability has been used in the study of classification and regression to derive generalization bounds [39, 40, 41, 42, 43, 44]. A key advantage of stability bounds is that they are designed for

specific learning algorithms, exploiting particular properties of the algorithms. Introduced 17 years ago, uniform stability [45] is now among the most widely used notions of algorithmic stability.

Given a training sample $\mathbf{S}$ of size $n$ and $i \in [n]$, remove the $i$-th element from $\mathbf{S}$, resulting in a sample of size $n-1$, which is denoted by $\mathbf{S}^{\backslash i} = ((x_1, y_1), \ldots, (x_{i-1}, y_{i-1}), (x_{i+1}, y_{i+1}) \ldots, (x_n, y_n))$. For a learning algorithm $\mathcal{A}$, define $f_{\mathbf{S}}^{\mathcal{A}} : \mathcal{X} \to \mathcal{Y}$ to be the the hypothesis that $\mathcal{A}$ has learned from the sample $\mathbf{S}$.

**Definition 4.1** (Uniform Stability [45])**.** *Given integer $n > 0$, the learning algorithm $\mathcal{A}$ is called $\beta_n$-uniformly stable with respect to the loss function $\ell$, if for any $i \in [n]$, $\mathbf{S} \in (\mathcal{X} \times \mathcal{Y})^n$, and $(x, y) \in \mathcal{X} \times \mathcal{Y}$, it holds that*

$$|\ell(y, f_{\mathbf{S}}^{\mathcal{A}}(x)) - \ell(y, f_{\mathbf{S}^{\backslash i}}^{\mathcal{A}}(x))| \leq \beta_n.$$

Intuitively, the stability of a leaning algorithm means that any small perturbation of training samples has little effect on the result of learning.

Now, we begin our analysis with studying the distribution of $\Phi_{\mathcal{A}}(\mathbf{S}) = R(f_{\mathbf{S}}^{\mathcal{A}}) - \widehat{R}(f_{\mathbf{S}}^{\mathcal{A}})$, namely, the difference between the empirical and the generalization errors. The mapping $\Phi_{\mathcal{A}} : (\mathcal{X} \times \mathcal{Y})^n \to \mathbb{R}$ will play a critical role in estimating $R(f_{\mathbf{S}}^{\mathcal{A}})$ via stability. We first show that the deviation of $\Phi_{\mathcal{A}}(\mathbf{S})$ from its expectation can be bounded with high probability (Lemma 4.2), and then upper bound the expected value of $\Phi_{\mathcal{A}}(\mathbf{S})$ in Lemma 4.3.

**Lemma 4.2.** *Given a sample $\mathbf{S}$ of size $n$ with dependency graph $G$, assume that the learning algorithm $\mathcal{A}$ is $\beta_n$-uniformly stable. Suppose the loss function $\ell$ is bounded by $M$. Then for any $t > 0$, it holds that*

$$\mathbf{Pr}(\Phi_{\mathcal{A}}(\mathbf{S}) - \mathbf{E}[\Phi_{\mathcal{A}}(\mathbf{S})] \geq t) \leq \exp\left(-\frac{2n^2 t^2}{\Lambda(G)(4n\beta_n + M)^2}\right).$$

Lemma 4.2 is proved in two steps. First, we treat $\Phi_{\mathcal{A}}(\cdot)$ as an $n$-ary function and show that its Lipschitz coefficients are all bounded by $4\beta_n + M/n$. Second, regarding $\mathbf{S}$ as a random vector, we apply Theorem 3.6 to $\Phi_{\mathcal{A}}(\mathbf{S})$. For detail, see Subsection B.1 of the supplementary materials.

**Lemma 4.3.** *Given a sample $\mathbf{S}$ of size $n$ with dependency graph $G$, assume that the learning algorithm $\mathcal{A}$ is $\beta_i$-uniformly stable for any $i \leq n$. Suppose the maximum degree of $G$ is $\Delta$. Let $\beta_{n,\Delta} = \max_{i \in [0,\Delta]} \beta_{n-i}$. It holds that*

$$\mathbf{E}[\Phi_{\mathcal{A}}(\mathbf{S})] \leq 2\beta_{n,\Delta}(\Delta + 1).$$

The proof of the lemma is based on iterative perturbations on the training sample $\mathbf{S}$. A perturbation is essentially removing a data point from or adding a data point to $\mathbf{S}$. The property of uniform stability of the algorithm guarantees that each perturbation causes a discrepancy up to $\beta_{n,\Delta}$, and in total $2(\Delta + 1)$ perturbations have to be made in order to *eliminate* the dependency between a data point and the others. For detail, please refer to Subsection B.2 of the supplementary materials.

Combining Lemma 4.2 and Lemma 4.3, we immediately have

**Theorem 4.4.** *Given a sample $\mathbf{S}$ of size $n$ with dependency graph $G$, assume that the learning algorithm $\mathcal{A}$ is $\beta_i$-uniformly stable for any $i \leq n$. Suppose the maximum degree $G$ is $\Delta$, and the loss function $\ell$ is bounded by $M$. Let $\beta_{n,\Delta} = \max_{i \in [0,\Delta]} \beta_{n-i}$. For any $\delta \in (0,1)$, with probability at least $1 - \delta$, it holds that*

$$R(f_{\mathbf{S}}^{\mathcal{A}}) \leq \widehat{R}(f_{\mathbf{S}}^{\mathcal{A}}) + 2\beta_{n,\Delta}(\Delta + 1) + \frac{4n\beta_n + M}{n}\sqrt{\frac{\Lambda(G)\ln(1/\delta)}{2}}.$$

**Remark 4.5.** It is well known that for many learning algorithms $\beta_n = O(1/n)$ [45]. Thus, we often have $\beta_{n,\Delta}(\Delta + 1) \leq \beta_{n-\Delta}(\Delta + 1) = O(\frac{\Delta}{n-\Delta})$, which vanishes asymptotically if $\Delta = o(n)$. The term $O\left(\sqrt{\Lambda(G)}/n\right)$ also vanishes asymptotically if $\Lambda(G) = o(n^2)$. As a result, in case of *weak* dependence such as the examples in Subsection 3.3, the generalization error is almost upper-bounded by the empirical error. We also observe that if the training data are i.i.d., Theorem 4.4 degenerates to the standard stability bound in [45], by applying $\Delta = 0$, $\beta_{n,\Delta} = \beta_n$, $\Lambda(G) = n$.

## 4.2 Application: Learning from $m$-dependent Data

We present a practical application in machine learning. Suppose there are linearly aligned locations, for example, real estates along a street. Let $y_i$ be the observation at location $i$, e.g., the house price, and $x_i$ stand for the random variable modeling geographical effect at location $i$. Suppose that $x$'s are mutually independent and each $y_i$ is geographically influenced by a neighborhood of size at most $2q+1$. One hope to learn the model of $y$ from a sample $\{((x_{i-q}, \ldots, x_i, \ldots, x_{i+q}), y_i)\}_{i \in [n]}$, where $n$ is the size of the sample. This model accounts for the impact of local locations on house prices. Similar scenarios are frequently considered in spatial econometrics, see [46] for more examples.

This application is a special case of $m$-dependence, which is an important statistical model introduced by Hoeffding in [47]. $m$-dependence has been studied extensively in probability, statistics, and combinatorics [48, 49, 50].

**Definition 4.6** ($m$-dependence [47]). *For some $m, n \in \mathbb{N}_+$, a sequence of random variables $\{X_i\}_{i=1}^n$ is called $m$-dependent if for any $i \in [n-m-1]$, $\{X_j\}_{j=1}^i$ is independent of $\{X_j\}_{j=i+m+1}^n$.*

The upper part of Figure 4 illustrates a dependency graph of 2-dependent sequence $\{X_i\}_{i=1}^n$.

As illustrated in Figure 4, we divide an $m$-dependent sequence into blocks of size $m$, and sequentially map the blocks to vertices of a path of length $\left\lceil \frac{n}{m} \right\rceil$. This forest approximation leads to

$$\Lambda(G) \leq \left( \left\lceil \frac{n}{m} \right\rceil - 1 \right) (m+m)^2 + m^2 \leq 4mn = O(mn)$$

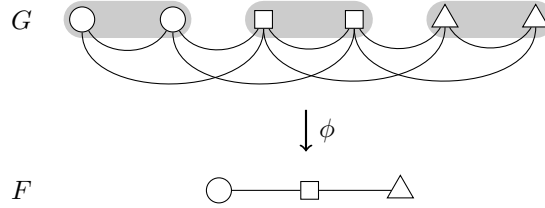

Figure 4: A forest approximation of a 2-dependent sequence. The approximation $\phi$ maps any vertex of $G$ to the vertex of $F$ having the same shape, so each gray belt stands for a pre-image set of $\phi$.

Combining Theorem 4.4 and the estimated forest complexity, we have

**Corollary 4.7.** *Given an $m$-dependent sequence $\mathbf{S}$ of length $n$ as training sample, assume that the learning algorithm $\mathcal{A}$ is $\beta_i$-uniformly stable for any $i \leq n$. Suppose the loss function $\ell$ is bounded by $M$. For any $\delta \in (0, 1)$, with probability at least $1 - \delta$, it holds that*

$$R(f_{\mathbf{S}}^{\mathcal{A}}) \leq \widehat{R}(f_{\mathbf{S}}^{\mathcal{A}}) + 2\beta_{n,2m}(2m+1) + (4n\beta_n + M)\sqrt{\frac{2m \ln(1/\delta)}{n}}.$$

Choose any uniformly stable learning algorithm $\mathcal{A}$ in [45] with $\beta_n = O(1/n)$, such as regularization algorithms in RKHS. Apply it to the above mentioned house price prediction problem. Then for any fixed $q$, with high probability, Corollary 4.7 leads to $R(f_{\mathbf{S}}^{\mathcal{A}}) \leq \widehat{R}(f_{\mathbf{S}}^{\mathcal{A}}) + O\left( \sqrt{\frac{\ln(1/\delta)}{n}} \right)$ for sufficiently large $n$, matching the stability bound of the i.i.d. case in [45].

## 5 Conclusion and Future Work

In this paper, we establish McDiarmid-type concentration inequalities for general functions of graph-dependent random variables. We apply our concentration results to obtain a stability-based generalization error bound for learning from graph-dependent samples. There are several possible extensions of this work.

- We provide upper bounds of the forest complexity for several classes of graphs. It is an interesting algorithmic problem to efficiently estimate the forest complexity. One heuristic method to do this on a connected graph is via graph diameter, by merging vertices of the same distances to a peripheral vertex, resulting in a path as long as the diameter.

- If more information of the dependency structure is known, e.g., dependency hypergraphs [24, 25], can we obtain better concentration inequalities and generalization bounds?

- In [3, 12, 6], generalization error is defined different from that in this paper. The differences between these two definitions has been discussed in [3, 12]. It is a natural question whether our results can be adapted to that definition.

- There are some newly introduced dependency graph models such as thresholded dependency graphs [51] and weighted dependency graphs [52, 53]. Can the problem in this paper be solved under these new models?

**Acknowledgments**

Rui (Ray) Zhang would like to thank Nick Wormald for valuable comments on an early version of this paper. Yuyi Wang would like to thank Ondřej Kuželka for very helpful discussions. Liwei Wang would like to thank Yunchang Yang for very helpful discussions. Xingwu Liu's work is partially supported by the National Key Research and Development Program of China (Grant No. 2016YFB1000201), the National Natural Science Foundation of China (61420106013), State Key Laboratory of Computer Architecture Open Fund (CARCH3410), and Youth Innovation Promotion Association of Chinese Academy of Sciences.

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
