[Supplementary Material]

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

)$ taking values in a product space $\mathbf{\Omega} = \prod_{i \in [n]} \Omega_i$. For any set $S \subseteq [n]$, we denote $\mathbf{X}_S = \{X_i\}_{i \in S}$, and $\mathbf{\Omega}_S = \prod_{i \in S} \Omega_i$ for convenience. The proof of Theorem 3.2 will rest on Lemma A.1, which intuitively means that the small deviation of

$$\mathbf{E}[f(\mathbf{X})|X_1 = x_1, \ldots, X_{i-1} = x_{i-1}, X_i = x_i]$$

with respect to $x_i$ for all $i$ leads to a high concentration of $f(\mathbf{X})$ around its expectation. Our task is thus reduced to show that when $x_1, \ldots, x_{i-1}$ is fixed,

$$\mathbf{E}[f(\mathbf{X})|X_1 = x_1, \ldots, X_{i-1} = x_{i-1}, X_i = x_i] - \mathbf{E}[f(\mathbf{X})|X_1 = x_1, \ldots, X_{i-1} = x_{i-1}, X_i = x_i']$$

is small for any $x_i$ and $x_i'$. This will be true due to Lemma A.4, if there is a good coupling, namely, jointly distributed variables $(\mathbf{Y}, \mathbf{Z})$ whose Hamming distance is small and whose marginals are the distributions of $\mathbf{X}$ conditional on $\{X_1 = x_1, \ldots, X_{i-1} = x_{i-1}, X_i = x_i\}$ and on $\{X_1 = x_1, \ldots, X_{i-1} = x_{i-1}, X_i = x_i'\}$, respectively. Hence, the main part of the proof is to construct such a coupling (Lemma A.3) whose feasibility relies on the strong independence among $\mathbf{X}$ (Lemma A.2). First of all, recall a lemma in literature.

**Lemma A.1** ([25]). *If for any $j \in [n]$ and $\mathbf{y} \in \mathbf{\Omega}_{[j-1]}$, there is $b_j \geq 0$ such that*

$$\sup_{\xi \in \Omega_j} \mathbf{E}[f(\mathbf{X})|\mathbf{X}_{[j-1]} = \mathbf{y}, X_j = \xi] - \inf_{\xi \in \Omega_j} \mathbf{E}[f(\mathbf{X})|\mathbf{X}_{[j-1]} = \mathbf{y}, X_j = \xi] \leq b_j \qquad (6)$$

*then for any $t > 0$,*

$$\mathbf{Pr}(f(\mathbf{X}) - \mathbf{E}[f(\mathbf{X})] \geq t) \leq \exp\left(-\frac{2t^2}{\sum_{j=1}^n b_j^2}\right).$$

By this lemma, it suffice to show that the small deviation of $\mathbf{E}[f(\mathbf{X})|X_1 = x_1, \ldots, X_{i-1} = x_{i-1}, X_i = x_i]$ with respect to $x_i$ for all $i$ is small to prove Theorem 3.2. Before continuing the proof, we assume that

**Well-rooted:** $G$ is rooted at the vertex $n$ and $c_n = c_{\min}$.

**Well-sorted:** For any $i, j \in V(G)$, $j$ is a descendent of $i$ only if $j < i$.

These assumptions will not lose generality, since we can relabel the variables $X_1, \ldots, X_n$ to meet the requirements.

For any non-root vertex $i \in V(G)$, let $p(i)$ be the parent vertex of $i$. For the rest of the section, arbitrarily fix $i \in [n]$ and define $S = [i+1, n] \setminus \{p(i)\}$, where $[j, k]$ stands for the set $\{j, \ldots, k\}$ of integers. Arbitrarily choose a vector $\mathbf{x} = (x_1, \ldots, x_n) \in \mathbf{\Omega}$ and an element $x_i' \in \Omega_i$. Let $\mathbf{x}' = (x_1, \ldots, x_{i-1}, x_i', x_{i+1}, \ldots, x_n)$. We have the following technical lemma, indicating that $\mathbf{X}_S$ is independent of $X_i$ if $\mathbf{X}_{[i-1]}$ is given.

**Lemma A.2.** *For any vector $\mathbf{y} \in \mathbf{\Omega}_S$,*

$$\mathbf{Pr}(\mathbf{X}_S = \mathbf{y}|\mathbf{X}_{[i]} = \mathbf{x}_{[i]}) = \mathbf{Pr}(\mathbf{X}_S = \mathbf{y}|\mathbf{X}_{[i]} = \mathbf{x}_{[i]}').$$

*Proof.* Let $T_i$ be the subtree of $G$ that is rooted at $i$. Our basic idea is to prove the stronger property that $\mathbf{X}_S$ is independent of the other parts of $\mathbf{X}_{[i]}$ if $\mathbf{X}_{[i-1] \setminus V(T_i)}$ is given. Since $[i] = V(T_i) \bigcup ([i-1] \setminus V(T_i))$, it suffices to show that $\mathbf{X}_{V(T_i)}$ is independent of $\{\mathbf{X}_S, \mathbf{X}_{[i-1] \setminus V(T_i)}\}$, which in turn is reduced to prove the following two claims due to the definition of the dependency graphs.

**Claim 1** : $N_G^+(T_i) \bigcap ([i-1] \setminus V(T_i)) = \emptyset$, where $N_G^+(T_i) = \bigcup_{k \in V(T_i)} N_G^+(k)$.

Proof of Claim 1: Arbitrarily choose $j \in [i-1] \setminus V(T_i)$. Suppose for contradiction that $j \in N_G(k)$ for some $k \in V(T_i)$, namely, $j$ is either a child or the parent of $k$. Since $j \notin V(T_i)$, we must have $j = p(k)$ and $k = i$, which implies $j > i$ due to the Assumption Well-rooted. A contradiction is reached, so $N_G(T_i) \bigcap ([i-1] \setminus V(T_i)) = \emptyset$. Because $N_G^+(T_i) \bigcap ([i-1] \setminus V(T_i)) = N_G(T_i) \bigcap ([i-1] \setminus V(T_i))$, Claim 1 holds.

**Claim 2** : $N_G^+(T_i) \bigcap S = \emptyset$.

Proof of Claim 2: Arbitrarily choose $j \in S = \{i+1, \ldots, n\} \setminus N_G(i)$. One immediately has $j \notin N_G^+(i)$. Suppose for contradiction that $j \in N_G^+(T_i)$. Then $j \in N_G^+(k)$ for some descendent $k$ of $i$, which means that either $j = i$ or $j$ is a descendent of $i$. This in turn means that $j \leq i$ due to the Assumption Well-sorted. A contradiction is reached, so Claim 2 holds.

Since $G$ is a dependency graph of $\mathbf{X}$, Claims 1 and 2 indicate that $\mathbf{X}_{V(T_i)}$ is independent of $\{\mathbf{X}_S, \mathbf{X}_{[i-1] \setminus V(T_i)}\}$. Then

$$
\begin{aligned}
& \mathbf{Pr}(\mathbf{X}_S = \mathbf{y} | X_j = x_j, j \in [i-1] \setminus V(T_i)) \\
= \ & \mathbf{Pr}(\mathbf{X}_S = \mathbf{y} | X_j = x_j, j \in ([i-1] \setminus V(T_i)) \bigcup V(T_i)) \\
= \ & \mathbf{Pr}(\mathbf{X}_S = \mathbf{y} | X_j = x_j, j \in [i]) \\
= \ & \mathbf{Pr}(\mathbf{X}_S = \mathbf{y} | \mathbf{X}_{[i]} = \mathbf{x}_{[i]}).
\end{aligned}
$$

Likewise, we also have

$$
\mathbf{Pr}(\mathbf{X}_S = \mathbf{y} | X_j = x'_j, j \in [i-1] \setminus V(T_i)) = \mathbf{Pr}(\mathbf{X}_S = \mathbf{y} | \mathbf{X}_{[i]} = \mathbf{x}'_{[i]})
$$

Since $\mathbf{x}$ and $\mathbf{x}'$ differ only in the $i$-th entry,

$$
\mathbf{Pr}(\mathbf{X}_S = \mathbf{y} | X_j = x_j, j \in [i-1] \setminus V(T_i)) = \mathbf{Pr}(\mathbf{X}_S = \mathbf{y} | X_j = x'_j, j \in [i-1] \setminus V(T_i)).
$$

As a result, $\mathbf{Pr}(\mathbf{X}_S = \mathbf{y} | \mathbf{X}_{[i]} = \mathbf{x}_{[i]}) = \mathbf{Pr}(\mathbf{X}_S = \mathbf{y} | \mathbf{X}_{[i]} = \mathbf{x}'_{[i]})$, this completes the proof of Lemma A.2. $\square$

Then we construct the jointly-distributed random vectors $(\mathbf{Y}, \mathbf{Z}) \in \Omega^2$ with respect to the fixed $i, \mathbf{x}$, and $\mathbf{x}'$. Specifically, $\mathbf{Y} = (Y_1, \ldots, Y_n)$ and $\mathbf{Z} = (Z_1, \ldots, Z_n)$ are defined as below.

1. $\mathbf{Y}_{[i]} = \mathbf{x}_{[i]}$
2. For any vector $\mathbf{y} \in \Omega_{[i+1,n]}$,

$$
\mathbf{Pr}(\mathbf{Y}_{[i+1,n]} = \mathbf{y}) = \mathbf{Pr}(\mathbf{X}_{[i+1,n]} = \mathbf{y} | \mathbf{X}_{[i]} = \mathbf{x}_{[i]})
$$

3. $\mathbf{Z}_{[i]} = \mathbf{x}'_{[i]}, \mathbf{Z}_S = \mathbf{Y}_S$.
4. For any vector $\mathbf{z} \in \Omega_S$ and element $z \in \Omega_{p(i)}$,

$$
\mathbf{Pr}(Z_{p(i)} = z | \mathbf{Z}_S = \mathbf{z}) = \mathbf{Pr}(X_{p(i)} = z | \mathbf{X}_{[i]} = \mathbf{x}'_{[i]}, \mathbf{X}_S = \mathbf{z})
$$

The next lemma states that $(\mathbf{Y}, \mathbf{Z})$ has the desired marginal distribution.

**Lemma A.3.** *For any vector $\mathbf{y} \in \Omega_{[i+1,n]}$, we have*

1. $\mathbf{Pr}(\mathbf{Y}_{[i+1,n]} = \mathbf{y}) = \mathbf{Pr}(\mathbf{X}_{[i+1,n]} = \mathbf{y} | \mathbf{X}_{[i]} = \mathbf{x}_{[i]})$,

2. $\mathbf{Pr}(\mathbf{Z}_{[i+1,n]} = \mathbf{y}) = \mathbf{Pr}(\mathbf{X}_{[i+1,n]} = \mathbf{y} | \mathbf{X}_{[i]} = \mathbf{x}'_{[i]})$.

*Proof.* (1) holds by the definition of $\mathbf{Y}$. To prove (2), arbitrarily choose $\mathbf{y} = (y_{i+1}, \ldots, y_n) \in \Omega_{[i+1,n]}$. Then we have

$$
\begin{aligned}
\mathbf{Pr}(\mathbf{Z}_{[i+1,n]} = \mathbf{y}) = \ & \mathbf{Pr}(\mathbf{Z}_S = \mathbf{y}_S) \mathbf{Pr}(Z_{p(i)} = y_{p(i)} | \mathbf{Z}_S = \mathbf{y}_S) \\
= \ & \mathbf{Pr}(\mathbf{Y}_S = \mathbf{y}_S) \mathbf{Pr}(Z_{p(i)} = y_{p(i)} | \mathbf{Z}_S = \mathbf{y}_S) \\
= \ & \mathbf{Pr}(\mathbf{X}_S = \mathbf{y}_S | \mathbf{X}_{[i]} = \mathbf{x}_{[i]}) \mathbf{Pr}(X_{p(i)} = y_{p(i)} | \mathbf{X}_{[i]} = \mathbf{x}'_{[i]}, \mathbf{X}_S = \mathbf{y}_S) \\
= \ & \mathbf{Pr}(\mathbf{X}_S = \mathbf{y}_S | \mathbf{X}_{[i]} = \mathbf{x}'_{[i]}) \mathbf{Pr}(X_{p(i)} = y_{p(i)} | \mathbf{X}_{[i]} = \mathbf{x}'_{[i]}, \mathbf{X}_S = \mathbf{y}_S) \\
= \ & \mathbf{Pr}(\mathbf{X}_{[i+1,n]} = \mathbf{y} | \mathbf{X}_{[i]} = \mathbf{x}'_{[i]}).
\end{aligned}
$$

where the fourth equality is due to Lemma A.2. $\square$

**Lemma A.4.** $\mathbf{E}[f(\mathbf{X}) | \mathbf{X}_{[i]} = \mathbf{x}_{[i]}] - \mathbf{E}[f(\mathbf{X}) | \mathbf{X}_{[i]} = \mathbf{x}'_{[i]}] \leq c_i + c_{p(i)}$.

*Proof.* By the definition of random vectors $\mathbf{Y}, \mathbf{Z}$ and Lemma A.3,

$$
\begin{aligned}
\mathbf{E}[f(\mathbf{X})|\mathbf{X}_{[i]} = \mathbf{x}_{[i]}] - \mathbf{E}[f(\mathbf{X})|\mathbf{X}_{[i]} = \mathbf{x}'_{[i]}] = \quad & \mathbf{E}[f(\mathbf{Y})] - \mathbf{E}[f(\mathbf{Z})] \\
= \quad & \mathbf{E}[f(\mathbf{Y}) - f(\mathbf{Z})] \\
\leq \quad & \mathbf{E}\left[\sum_{j=1}^{n} c_j \mathbf{1}_{Y_j \neq Z_j}\right] \\
\leq \quad & c_i + c_{p(i)}.
\end{aligned}
$$

the first equality is due to the coupling constructed before, and the first inequality is by triangle inequality and $\mathbf{c}$-Lipschitz properties of $f$. $\square$

We are now ready to prove Theorem 3.2.

*Proof of Theorem 3.2.* By Lemma A.1 and Lemma A.4

$$
\begin{aligned}
\mathbf{Pr}(f(\mathbf{X}) - \mathbf{E}[f(\mathbf{X})] \geq t) \leq \quad & \exp\left(-\frac{2t^2}{\sum_{j \in V(G) \setminus \{n\}} (c_j + c_{p(j)})^2 + c_n^2}\right) \\
= \quad & \exp\left(-\frac{2t^2}{\sum_{\langle j,k \rangle \in E(G)} (c_j + c_k)^2 + c_{\min}^2}\right)
\end{aligned}
$$

the last equality is because the root $n$ has no parent and the **Well-rooted** assumption. $\square$

## A.2 Proof of Theorem 3.3

*Proof of Theorem 3.3.* The proof is similar to that of Theorem 3.2. Without loss of generality, we assume that each component of the forest $G$ are well-rooted and well-sorted. Then the proofs of Lemma A.2-A.4 remain valid, since variables in different components are independent. As a result, the theorem holds due to Lemma A.1.

$\square$

## A.3 Proof of Theorem 3.6

**Lemma A.5.** *Suppose that $f : \Omega \to \mathbb{R}$ is a $\mathbf{c}$-Lipschitz function and $G$ is a dependency graph of a random vector $\mathbf{X}$ that takes values in $\Omega$. For any $t > 0$ and any $(\phi, F) \in \Phi(G)$ with $F$ consisting of trees $\{T_i\}_{i \in [k]}$, the following inequality holds:*

$$
\mathbf{Pr}(f(\mathbf{X}) - \mathbf{E}[f(\mathbf{X})] \geq t) \leq \exp\left(-\frac{2t^2}{\sum_{\langle u,v \rangle \in E(F)} (\widetilde{c}_u + \widetilde{c}_v)^2 + \sum_{i=1}^{k} \widetilde{c}_{\min,i}^2}\right)
$$

*where $\widetilde{c}_u = \sum_{i \in \phi^{-1}(u)} c_i$ and $\widetilde{c}_{\min,i} = \min_{u \in V(T_i)} \widetilde{c}_u$. Here, $\phi^{-1}(u)$ is the set of pre-images of $u$.*

*Proof.* For any $u \in V(F)$, define a random vector $\mathbf{Y}_u = \{X_i\}_{i \in \phi^{-1}(u)}$. Treat each $\mathbf{Y}_u$ as a random variable. Define a new random vector $\mathbf{Y} = (\mathbf{Y}_u)_{u \in V(F)}$, and let $g(\mathbf{Y}) = f(\mathbf{X})$. It is easy to check that $g$ is $\widetilde{\mathbf{c}}$-Lipschitz, where $\widetilde{\mathbf{c}} = (\widetilde{c}_u)_{u \in V(F)}$. The theorem immediately follows from Theorem 3.3. $\square$

Lemma A.5 immediately implies Theorem 3.6 by the definition of forest complexity.

# B Omitted Proofs in Section 4

## B.1 Proof of Lemma 4.2

The following technical lemma is needed.

**Lemma B.1** ([43]). *Given a $\beta_n$-uniformly stable learning algorithm $\mathcal{A}$, for any $\mathbf{S}, \mathbf{S}' \in (\mathcal{X} \times \mathcal{Y})^n$ that differ only in one entry, it holds that*

$$
|\Phi_{\mathcal{A}}(\mathbf{S}) - \Phi_{\mathcal{A}}(\mathbf{S}')| \leq 4\beta_n + \frac{M}{n}
$$

*Proof.* In the literature, Lemma B.1 was proved for i.i.d. data, actually, the proof remains valid in our setting. Assume $\mathbf{S}$, $\mathbf{S}'$ differ only in $i$-th entry, and denote $\mathbf{S}'$ as $\mathbf{S}^i$

$$\mathbf{S}^i = ((x_1, y_1), \ldots, (x_{i-1}, y_{i-1}), (x_i', y_i'), (x_{i+1}, y_{i+1}) \ldots, (x_m, y_m))$$

and the marginal distribution of $(x_i', y_i')$ is also $D$.

Notice that we do not require the data to be i.i.d., samples are dependent with the same marginal probability distribution $D$. First, we bound $R(f_{\mathbf{S}}^{\mathcal{A}}) - R(f_{\mathbf{S}^i}^{\mathcal{A}})$

$$|R(f_{\mathbf{S}}^{\mathcal{A}}) - R(f_{\mathbf{S}^i}^{\mathcal{A}})| \tag{7}$$

$$\leq |R(f_{\mathbf{S}}^{\mathcal{A}}) - R(f_{\mathbf{S}\setminus i}^{\mathcal{A}})| + |R(f_{\mathbf{S}\setminus i}^{\mathcal{A}}) - R(f_{\mathbf{S}^i}^{\mathcal{A}})| \tag{8}$$

$$= |\mathbf{E}_D[\ell(y, f_{\mathbf{S}}^{\mathcal{A}}(x))] - \mathbf{E}_D[\ell(y, f_{\mathbf{S}\setminus i}^{\mathcal{A}}(x))]| + |\mathbf{E}_D[\ell(y, f_{\mathbf{S}\setminus i}^{\mathcal{A}}(x))] - \mathbf{E}_D[\ell(y, f_{\mathbf{S}^i}^{\mathcal{A}}(x))]| \tag{9}$$

$$= |\mathbf{E}_D[\ell(y, f_{\mathbf{S}}^{\mathcal{A}}(x)) - \ell(y, f_{\mathbf{S}\setminus i}^{\mathcal{A}}(x))]| + |\mathbf{E}_D[\ell(y, f_{\mathbf{S}\setminus i}^{\mathcal{A}}(x)) - \ell(y, f_{\mathbf{S}^i}^{\mathcal{A}}(x))]| \tag{10}$$

$$\leq 2\beta_n \tag{11}$$

then, we bound $\widehat{R}(f_{\mathbf{S}}^{\mathcal{A}}) - \widehat{R}_{\mathbf{S}^i}(f_{\mathbf{S}^i}^{\mathcal{A}})$

$$n|\widehat{R}(f_{\mathbf{S}}^{\mathcal{A}}) - \widehat{R}_{\mathbf{S}^i}(f_{\mathbf{S}^i}^{\mathcal{A}})| \tag{12}$$

$$= \left| \sum_{(x_j, y_j) \in \mathbf{S}} \ell(y_j, f_{\mathbf{S}}^{\mathcal{A}}(x_j)) - \sum_{(x_j, y_j) \in \mathbf{S}^i} \ell(y_j, f_{\mathbf{S}^i}^{\mathcal{A}}(x_j)) \right| \tag{13}$$

$$\leq \sum_{j \neq i} |\ell(y_j, f_{\mathbf{S}}^{\mathcal{A}}(x_j)) - \ell(y_j, f_{\mathbf{S}^i}^{\mathcal{A}}(x_j))| + |\ell(y_i, f_{\mathbf{S}}^{\mathcal{A}}(x_i)) - \ell(y_i', f_{\mathbf{S}^i}^{\mathcal{A}}(x_i'))| \tag{14}$$

$$\leq \sum_{j \neq i} |\ell(y_j, f_{\mathbf{S}}^{\mathcal{A}}(x_j)) - \ell(y_j, f_{\mathbf{S}\setminus i}^{\mathcal{A}}(x_j))| + \sum_{j \neq i} |\ell(y_j, f_{\mathbf{S}\setminus i}^{\mathcal{A}}(x_j)) - \ell(y_j, f_{\mathbf{S}^i}^{\mathcal{A}}(x_j))|$$

$$+ |\ell(y_i, f_{\mathbf{S}}^{\mathcal{A}}(x_i)) - \ell(y_i', f_{\mathbf{S}^i}^{\mathcal{A}}(x_i'))| \tag{15}$$

$$\leq 2n\beta_n + M \tag{16}$$

combining above bounds, we have

$$\begin{aligned} |\Phi_{\mathcal{A}}(\mathbf{S}) - \Phi_{\mathcal{A}}(\mathbf{S}^i)| &= |(R(f_{\mathbf{S}}^{\mathcal{A}}) - \widehat{R}(f_{\mathbf{S}}^{\mathcal{A}})) - (R(f_{\mathbf{S}^i}^{\mathcal{A}}) - \widehat{R}_{\mathbf{S}^i}(f_{\mathbf{S}^i}^{\mathcal{A}}))| \\ &\leq |R(f_{\mathbf{S}}^{\mathcal{A}}) - R(f_{\mathbf{S}^i}^{\mathcal{A}})| + |\widehat{R}(f_{\mathbf{S}}^{\mathcal{A}}) - \widehat{R}_{\mathbf{S}^i}(f_{\mathbf{S}^i}^{\mathcal{A}})| \\ &\leq 4\beta_n + \frac{M}{n} \end{aligned}$$

$\square$

combining Lemma B.1 and Theorem 3.6 leads to Lemma 4.2.

## B.2 Proof of Lemma 4.3

We introduce a technical lemma before the proof of Lemma 4.3.

**Lemma B.2.** *Given a sample $\mathbf{S}$ of size $n$ with dependency graph $G$, assume that the learning algorithm $\mathcal{A}$ is $\beta_i$-uniformly stable for any $i \leq n$. Suppose the maximum degree of $G$ is $\Delta$. Let $\beta_{n,\Delta} = \max_{i \in [0, \Delta]} \beta_{n-i}$. It holds that*

$$\max_{(x_i, y_i) \in \mathbf{S}} \mathbf{E}_{\mathbf{S}, (x, y)}[\ell(y, f_{\mathbf{S}}^{\mathcal{A}}(x)) - \ell(y_i, f_{\mathbf{S}}^{\mathcal{A}}(x_i))] \leq 2\beta_{n,\Delta}(\Delta + 1).$$

*Proof.* For any $i \in [n]$, suppose $N_G^+(i) = \{j_1, \ldots, j_{n_i}\}$ with $j_{k-1} > j_k$. Define $\mathbf{S}^{(i,0)} = \mathbf{S}$ and for $k \in [n_i]$, $\mathbf{S}^{(i,k)}$ is obtained from $\mathbf{S}^{(i,k-1)}$ by removing the $j_k$-th entry. By uniform stability of $\mathcal{A}$, for any $(x, y) \in \mathcal{X} \times \mathcal{Y}$ and $k \in [n_i]$,

$$|\ell(y, f_{\mathbf{S}^{(i,k-1)}}^{\mathcal{A}}(x)) - \ell(y, f_{\mathbf{S}^{(i,k)}}^{\mathcal{A}}(x))| \leq \beta_{n,\Delta}$$

we have the decomposition via telescoping

$$\ell(y, f_{\mathbf{S}}^{\mathcal{A}}(x)) = \sum_{k=1}^{n_i} (\ell(y, f_{\mathbf{S}^{(i,k-1)}}^{\mathcal{A}}(x)) - \ell(y, f_{\mathbf{S}^{(i,k)}}^{\mathcal{A}}(x)) + \ell(y, f_{\mathbf{S}^{(i,n_i)}}^{\mathcal{A}}(x))$$

similarly

$$\ell(y_i, f_{\mathbf{S}}^{\mathcal{A}}(x_i)) = \sum_{k=1}^{n_i} (\ell(y_i, f_{\mathbf{S}^{(i,k-1)}}^{\mathcal{A}}(x_i)) - \ell(y_i, f_{\mathbf{S}^{(i,k)}}^{\mathcal{A}}(x_i)) + \ell(y_i, f_{\mathbf{S}^{(i,n_i)}}^{\mathcal{A}}(x_i))$$

Thus, we have

$$
\begin{aligned}
& \ell(y, f_{\mathbf{S}}^{\mathcal{A}}(x)) - \ell(y_i, f_{\mathbf{S}}^{\mathcal{A}}(x_i)) \\
=\ & \sum_{k=1}^{n_i} \left( (\ell(y, f_{\mathbf{S}^{(i,k-1)}}^{\mathcal{A}}(x)) - \ell(y, f_{\mathbf{S}^{(i,k)}}^{\mathcal{A}}(x))) - (\ell(y_i, f_{\mathbf{S}^{(i,k)}}^{\mathcal{A}}(x_i)) - \ell(y_i, f_{\mathbf{S}^{(i,k-1)}}^{\mathcal{A}}(x_i))) \right) \\
& + \ell(y, f_{\mathbf{S}^{(i,n_i)}}^{\mathcal{A}}(x)) - \ell(y_i, f_{\mathbf{S}^{(i,n_i)}}^{\mathcal{A}}(x_i)) \\
\leq\ & \sum_{k=1}^{n_i} |\ell(y, f_{\mathbf{S}^{(i,k-1)}}^{\mathcal{A}}(x)) - \ell(y, f_{\mathbf{S}^{(i,k)}}^{\mathcal{A}}(x))| \\
& + \sum_{k=1}^{n_i} |\ell(y_i, f_{\mathbf{S}^{(i,k)}}^{\mathcal{A}}(x_i)) - \ell(y_i, f_{\mathbf{S}^{(i,k-1)}}^{\mathcal{A}}(x_i))| + \ell(y, f_{\mathbf{S}^{(i,n_i)}}^{\mathcal{A}}(x)) - \ell(y_i, f_{\mathbf{S}^{(i,n_i)}}^{\mathcal{A}}(x_i)) \\
\leq\ & 2n_i \beta_{n,\Delta} + \ell(y, f_{\mathbf{S}^{(i,n_i)}}^{\mathcal{A}}(x)) - \ell(y_i, f_{\mathbf{S}^{(i,n_i)}}^{\mathcal{A}}(x_i))
\end{aligned}
$$

As a result,

$$
\begin{aligned}
& \mathbf{E}_{\mathbf{S},(x,y)}[\ell(y, f_{\mathbf{S}}^{\mathcal{A}}(x)) - \ell(y_i, f_{\mathbf{S}}^{\mathcal{A}}(x_i))] \\
=\ & \mathbf{E}_{\mathbf{S},(x,y)}[\ell(y, f_{\mathbf{S}^{(i,n_i)}}^{\mathcal{A}}(x)) - \ell(y_i, f_{\mathbf{S}^{(i,n_i)}}^{\mathcal{A}}(x_i))] + 2n_i \beta_{n,\Delta} \\
\leq\ & \mathbf{E}_{\mathbf{S},(x,y)}[\ell(y, f_{\mathbf{S}^{(i,n_i)}}^{\mathcal{A}}(x)) - \ell(y_i, f_{\mathbf{S}^{(i,n_i)}}^{\mathcal{A}}(x_i))] + 2\beta_{n,\Delta}(\Delta + 1) \\
=\ & \mathbf{E}_{\mathbf{S},(x,y)}[\ell(y, f_{\mathbf{S}^{(i,n_i)}}^{\mathcal{A}}(x))] - \mathbf{E}_{\mathbf{S}}[\ell(y_i, f_{\mathbf{S}^{(i,n_i)}}^{\mathcal{A}}(x_i))] + 2\beta_{n,\Delta}(\Delta + 1) \\
=\ & \mathbf{E}_{\mathbf{S}^{(i,n_i)},(x,y)}[\ell(y, f_{\mathbf{S}^{(i,n_i)}}^{\mathcal{A}}(x))] - \mathbf{E}_{\mathbf{S}^{(i,n_i)},(x_i,y_i)}[\ell(y_i, f_{\mathbf{S}^{(i,n_i)}}^{\mathcal{A}}(x_i))] + 2\beta_{n,\Delta}(\Delta + 1) \\
=\ & 2\beta_{n,\Delta}(\Delta + 1)
\end{aligned}
$$

The last equality is because $(x_i, y_i)$ and $(x, y)$ are independent of $\mathbf{S}^{(i,n_i)}$ and have the same distribution.  $\square$

*Proof of Lemma 4.3.*

$$
\begin{aligned}
\mathbf{E}_{\mathbf{S}}[\Phi_{\mathcal{A}}(\mathbf{S})] =\ & \mathbf{E}_{\mathbf{S}}[\mathbf{E}_{(x,y)}[\ell(y, f_{\mathbf{S}}^{\mathcal{A}}(x))] - \frac{1}{n}\sum_{i=1}^{n} \ell(y_i, f_{\mathbf{S}}^{\mathcal{A}}(x_i))] \\
=\ & \frac{1}{n}\sum_{i=1}^{n} \mathbf{E}_{\mathbf{S},(x,y)}[\ell(y, f_{\mathbf{S}}^{\mathcal{A}}(x)) - \ell(y_i, f_{\mathbf{S}}^{\mathcal{A}}(x_i))] \\
\leq\ & 2\beta_{n,\Delta}(\Delta + 1)
\end{aligned}
$$

$\square$