[Reviews · NeurIPS 2019]

Reviewer 1



Background ======== The paper considers the important challenge of generalization theory for dependent variables. Dependency graphs provide a natural and convenient method for capturing such dependencies where two subsets of RV are independent iff they are non-adjacent. Several previous results included generalization of Hoeffding's inequality that is useful in the context of bounding the deviation of the average of (dependent) RV from their mean. Main result ======= McDiarmid-type are more general and allow analyzing the deviation of any Lipschitz function from its mean. The main contribution paper is a McDiarmid inequality for graph dependent RV in terms of the Lipschitz constants corresponding to the dependent RV. Application ======== Stability bounds provide a great motivation for McDiarmid-type inequalities. The authors suggest high probability stability bounds for dependent variables and illustrate the applicability for learning in m-dependent data. Overall I find the contribution significant and interesting.

Reviewer 2



I read about 75% of the paper and what I read seems to be correct.

Reviewer 3



This is a solid paper which addresses an important problem in a novel way. It is very well written, as the Authors guide the reader through the presentation of novel concepts, and they do a good job at easing the reader into the subject with a good review of existing works. The results are very interesting and they should have significant impact. While I haven't checked all the proofs, the reasoning seems correct, and the results are in line with the intuition.

[Author Response · NeurIPS 2019]

We thank all reviewers for their efforts towards improving our manuscript. In the following paragraphs, we mainly address the concerns of Reviewer #3.

**Comment #1.** *Give a clear example of forest approximation. The brief paragraph on merging nodes was not very clear.*

**Response.** By definition, the forest approximation of a graph $G$ by a forest $F$ is a mapping from vertices of $G$ to $F$ such that the images of any pair of adjacent vertices are either the same or adjacent.

Recall Figure 2 in our manuscript, which is presented below in Figure 1. $G$ is a cycle of 5 vertices and $F$ is a path of 3 vertices. $\phi$ maps each vertex of $G$ to the vertex of $F$ which has the same shape. Say, vertices 1 and 2 of $G$ are mapped to vertex $u$ of $F$, likewise for the other vertices. We can check that the images of any pair of adjacent vertices of $G$ are either the same or adjacent in $F$. For example, vertices 1 and 2 have the same image $u$, while 3 and 4 are mapped to $v$ and $w$ respectively, which are adjacent in $F$. As a result, $\phi$ is a forest approximation of $G$ by $F$.

Figure 1: A forest approximation of $C_5$

[Meta-Review · NeurIPS 2019]

This paper presents a new theoretical concentration type inequality for dependence graphs. The reviewers found the paper solid and the results were very interesting.